# Islands of Confidence: Robust Neural Network Classification with Uncertainty Quantification

## Abstract

We propose a Gaussian confidence measure and its optimization, for use in neural network classifiers. The measure comes with theoretical results, simultaneously resolving two pressing problems in Deep Neural Network classification: uncertainty quantification, and robustness. Existing research in uncertainty quantification mostly revolves around the confidence reflected in the input feature space. Instead, we focus on the learned representation of the network and analyze the confidence in the penultimate layer space. We formally prove that, independent of optimization-procedural effects, a set of centroids always exists such that softmax classifiers are nearest-centroid classifiers. Softmax confidence, however, does not reflect that the classification is based on nearest centroids: artificially inflated confidence is also given to out-of-distributions samples that are not near any centroid, but slightly less distant from one centroid than from the others. Our new confidence measure is centroid-based, and hence no longer suffers from the artificial confidence inflation of out-of-distribution samples. We also show that our proposed centroidal confidence measure is providing a robustness certificate against attacks. As such, it manages to reflect what the model doesn't know (as demanded by uncertainty quantification), and to resolve the issue of robustness of neural networks.

## 1 Introduction

The last layer of state-of-the-art neural networks computes the final classifications by approximation through the softmax function (Boltzmann, 1868). This function partitions the transformed input space into Voronoi calls, each of which encompasses a single class. Conceptually, this is equivalent to putting a number of centroids in this transformed space, and clustering the data points in the dataset by proximity to these centroids through $k$-means. Several recent papers posed that exploring a relation between softmax and $k$-means can be beneficial (Kilinc & Uysal, 2018; Peng et al., 2018; Schilling et al., 2018).

The current state of scientific knowledge on the relation between $k$-means and softmax is empirical. In this paper, we theoretically prove that softmax is a centroid-based classifier, and we derive a centroid-based robustness certificate. This certificate motivates the usage of a confidence measure[1], the *Gauss confidence*, which reflects the distance of observations to their assigned centroids. Gauss confidence therefore expresses the uncertainties of the model; moreover, it indicates the vulnerabilities to attacks.

We show that our Gauss networks can match (median absolute difference: 0.45 percentage points) the test accuracy of softmax networks, but at a lower confidence (as desired); both outperform the competing DUQ networks (van Amersfoort et al., 2020) when the dataset has many classes. The lower confidence also results in Gauss networks being much less susceptible to adversarial attacks. Hence, the islands of confidence as illustrated in the rightmost plot of Figure 1 reflect reality much better than the confidence landscapes of existing methods (cf. other two plots in Figure 1).

---

[1]previously published informally (non-peer-reviewed) in January 2020 as (NN et al., 2020)

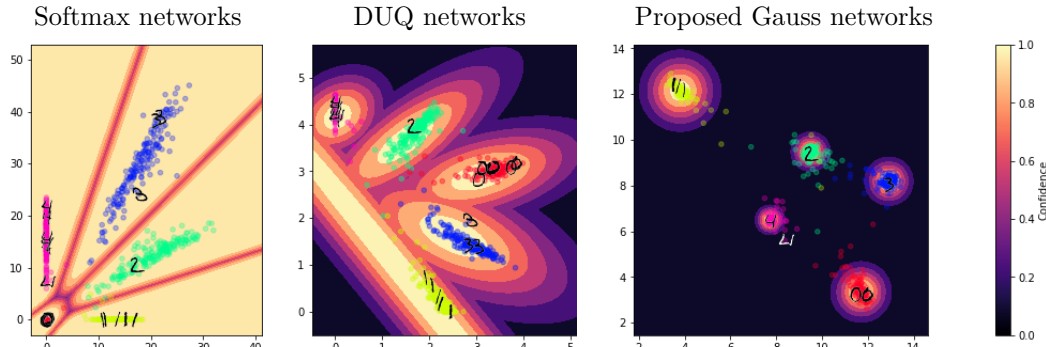

Figure 1: Classification confidence plots of actual decision boundaries/areas of a $d = 2$-dimensional penultimate layer space for three distinct Deep Neural Networks. The hand-written numbers indicate the MNIST class predictions in $\{0, \dots, 4\}$.

## 2 Related Work

Peng et al. (2018); Kilinc & Uysal (2018) were among the first to observe that the penultimate layer has surprising clustering properties. The phenomenon that samples gather around their centroid in the transformed feature space recently gained popularity under the term *neural collapse* (Kothapalli et al., 2022; Papyan et al., 2020). This motivates an adaptation of the confidence measure. Softmax confidence has the well-documented issue that it extrapolates with unjustified high confidence data points which are very far from the training data (Gal, 2016): softmax is overconfident. The consequences are observable in the very confident classification of noisy, entirely randomly generated images (Nguyen et al., 2015), and the effect that softmax probabilities are badly calibrated (Guo et al., 2017). Figure 1 illustrates that this effect is not surprising: in the transformed feature space of the penultimate layer, points that do not lie directly on the decision boundaries are all confidently assigned to a class. Softmax confidence has no capability to express what the model doesn't know.

### 2.1 Quantifying Uncertainties of DNNs

To remedy the *softmax issue*, various methods have been proposed. Although networks with a centroid-based confidence intuitively make sense, training them is a major challenge. When using the softmax cross-entropy loss, pushing a transformed feature vector $\phi(x)$ away from one class region means that it is pushed towards other class regions. Employing a centroidal confidence, one could push a point away from all centroids. This introduces trivial global optima of common loss functions, which does not lead to a well-performing classifier. To this end, Wen et al. (2016); Pang et al. (2019) minimize the distance of transformed feature points to given (fixed) centroids. Wan et al. (2018) employ a centroidal cross entropy loss, which directly optimizes for a nearest centroid classifier, but which does not increase the density around centroids. Likewise, Lebedev et al. (2018) define a confidence measure based on the normalized RBF kernel function. Hobbhahn et al. (2022); Mukhoti et al. (2021) propose a retraining of only the last layer to reflect uncertainties based on a multivariate Gaussian representation of confidences. Closest to our approach are Deterministic Uncertainty Classifiers (DUQs) (van Amersfoort et al., 2020), which directly optimize for a multivariate Gaussian confidence measure (cf. Figure 1). The authors employ binary cross-entropy loss, which is given for one-hot encoded labels $Y^* \in \mathbb{R}^{m \times c}$ for $m$ samples and $c$ classes by

$$\ell_{\text{BCE}}(f_{\text{DUQ}}, Y^*) = -\frac{1}{mc} \sum_{j=1}^{m} \sum_{k=1}^{c} Y_{jk}^* \log\left(f_{\text{DUQ}}(\mathbf{x}_j)\right) + (1 - Y_{jk}^*) \log\left(1 - f_{\text{DUQ}}(\mathbf{x}_j)\right)$$

$$f_{\text{DUQ}}(\mathbf{x})_k = \exp\left(-(\phi(\mathbf{x}) - Z_{\cdot k})^\top \Sigma_k^{-1} (\phi(\mathbf{x}) - Z_{\cdot k})\right).$$

Here, $f_{\mathrm{DUQ}} : \mathbb{R}^n \to [0,1]^c$ returns the prediction confidences, $Z \in \mathbb{R}^{d \times c}$ gathers the centroids by its columns and $\Sigma_k \in \mathbb{R}^{d \times d}$ is a scaled covariance matrix. BCE loss works well for data with a moderate number of classes. With increasing class complexity, DUQ models converge to the global optimum where all training data is infinitely far from all centroids ($\|\Sigma^{-1}\| \to 0$).

Related to methods that use RBF- or Gaussian mixture-based uncertainty measures are DUE (van Amersfoort et al., 2021) and SNGP (Liu et al., 2022), which train a network with a final Gaussian processes layer. Yet, also other distributions are suitable to reflect uncertainties. For example, prior networks (Malinin & Gales, 2018; 2019) reflect the training data domain via a Dirichlet distribution.

## 2.2 Robustness of Deep Neural Networks (DNNs)

The existence of adversarial examples demonstrates that SOTA deep learning models have warped inner representations of classes/object properties. The fact that adding humanly indistinguishable noise changes the classification from a bus to an ostrich is worrying. Consequently, a lot of research focuses on performing and preventing these attacks. The result is a circular development of attacks and defenses (Akhtar & Mian, 2018; Carlini et al., 2019; Athalye et al., 2018), which creates a need for certifiable robustness. A certificate provides, e.g., a lower bound on the magnitude of adversarial perturbation required to change the class. Existing certificates rely on bounds of the *global* Lipschitz constant, limiting the effect of any, arbitrarily large perturbation (Szegedy et al., 2014; Qian & Wegman, 2019; Gouk et al., 2018). Experimental and theoretical evaluations show that global Lipschitz bounds overly curtail the expressiveness of DNNs, yielding underfitting classifiers (Huster et al., 2018). The expressiveness of DNNs, their ability to approximate almost any function (Lu et al., 2017), is an important property which distinguishes feature transformations learned by DNNs from popular alternatives. Hence, we require a control of the *local* Lipschitz constant (restricting only the effect of small perturbations). Unfortunately, this is currently only possible for DNNs having one hidden layer (Hein & Andriushchenko, 2017).

Many uncertainty quantification methods regularize towards bi-Lipschitz functions. Bi-Lipschitz functions preserve distances in the output to a degree, that is specified by the Lipschitz constant. This prevents that all data (also out-of-distribution data) is mapped onto the same confidently assigned regions. DUQ uses gradient penalization, and DUE, SGNP, and DDU use spectral normalization (Gouk et al., 2021). Unfortunately, gradient penalization is slow and it only bounds the local Lipschitz constant for the data points it is optimized on. In comparison, spectral normalization is faster. However, applying spectral normalization such that it doesn't notably hurt the performance does not bound the Lipschitz constant enough to provide relevant robustness guarantees.

Another way to increase robustness is the maximum margin approach. Enlarging the margin between classes in the transformed feature space should require larger perturbances to push an example over the decision boundary (Liu et al., 2016; Pang et al., 2019). The success of this approach still depends on the Lipschitz constant of the feature transformation, requiring controlling the Lipschitz constant while maintaining expressiveness (Tsuzuku et al., 2018). With Gauss networks we show that a centroidal confidence measure provides a guarantee on the robustness which depends on both the Lipschitz constant and the margin.

## 3 Why Softmax Is Actually a Centroid-based Classifier

Consider a feedforward network, mapping points in $n$-dimensional space to a $c$-dimensional probability vector (with $c$ the number of classes), computing predictions by the function

$$f_{\mathrm{sm}}(\mathbf{x}) = \mathrm{softmax}\left(W^\top \phi(\mathbf{x}) + \mathbf{b}\right).$$

Here, $\phi(\mathbf{x}) \in \mathbb{R}^d$ returns the output of the penultimate layer. The last layer is linear; its weights are represented by the matrix $W \in \mathbb{R}^{d \times c}$ and bias $\mathbf{b} \in \mathbb{R}^c$.

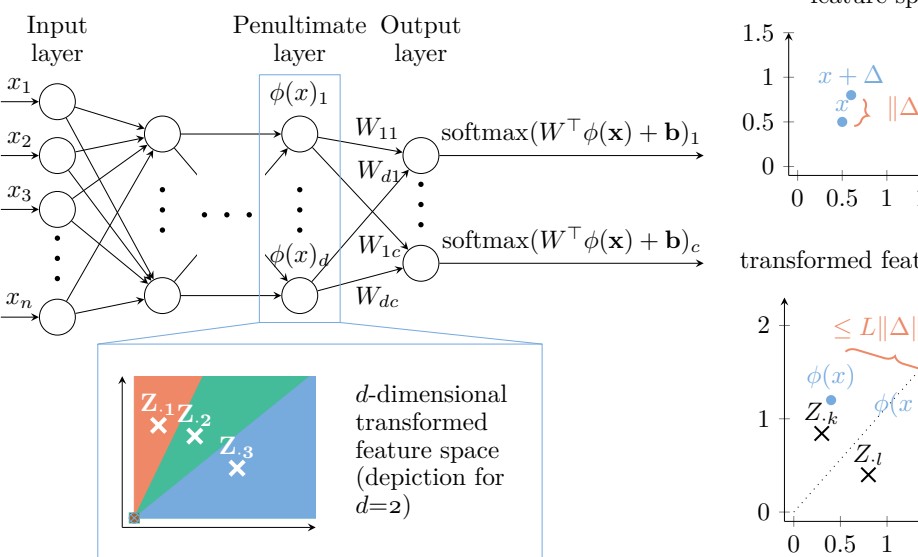

(a) A feed-forward neural network, computing a feature transformation $\phi$ and the representation of classes in the transformed feature space by convex cones, induced by centroids $Z_{\cdot 1}, \ldots, Z_{\cdot c}$.

(b) Gauss confidence certifies robustness: a minimum distance must be bridged to change the predicted class.

Figure 2: The role of Gauss confidence for Deep Neural Networks.

### 3.1 Softmax and Inherent Clustering Properties

Let $\mathcal{D} = \{(\mathbf{x}_1, y_1), \ldots, (\mathbf{x}_m, y_m)\}$ denote our training data of data points $\mathbf{x}_j \in \mathbb{R}^n$ and their corresponding class labels $y_j \in \{1, \ldots, c\}$. We denote by $\|\cdot\|$ the vector $l_2$-norm or the matrix Frobenius norm and by $|\cdot|$ the vector $l_1$-norm. Figure 2a illustrates the result of Theorem 1, showing that softmax-based classifiers are nearest centroid classifiers.

**Theorem 1.** *Let $W \in \mathbb{R}^{d \times c}$ be the matrix of weights between the penultimate layer and the output layer. Let $\phi : \mathbb{R}^n \to \mathbb{R}^d$ be the DNN output of the penultimate layer. If the network computes predictions over the (soft)max function with bias term $\mathbf{b}$ such that $y = \arg\max_k \phi(\mathbf{x})^\top W_{\cdot k} + b_k$ and if $W$ has at least a rank of $r \geq c$, then there exist $c$ class centroids $Z_{\cdot k} \in \mathbb{R}^d$ such that every point $\mathbf{x}$ is assigned to the class having the nearest centroid:*

$$y = \arg\min_k \|\phi(\mathbf{x}) - Z_{\cdot k}\|^2.$$

*Proof.* We show that for any dataset and network there exists a set of centroids such that the classification does not change when classifying according to the nearest centroid.

We gather the outputs of the penultimate layer in the matrix $D$ reflecting the transformed feature vectors as rows, or conversely, as columns in $D^\top$:

$$D^\top = (\phi(\mathbf{x}_1) \quad \ldots \quad \phi(\mathbf{x}_m)) \in \mathbb{R}^{d \times m}.$$

We define $Z = W + \mathbf{v}\mathbf{1}_c^\top$, where $\mathbf{v} \in \mathbb{R}^d$ and $\mathbf{1}_c \in \{1\}^c$ is a constant one vector. The (soft)max classification of all data points in $\mathcal{D}$ is then given by the one-hot encoded matrix $Y \in \{0, 1\}^{m \times c}$ that optimizes the objective

$$\arg\max_Y \operatorname{tr}(Y(W^\top D^\top + \mathbf{b}\mathbf{1}_m^\top)) = \arg\min_Y \|D - YZ^\top\|^2 + \operatorname{tr}((2\mathbf{b}\mathbf{1}_c^\top - Z^\top Z)Y^\top Y)) \quad (1)$$

Full derivation of this equivalence is given in Appendix A. The matrix $Z \in \mathbb{R}^{d \times c}$ indicates a set of $c$ centroids by its columns. The first term of Equation (1) is minimized if $Y$ assigns

the class with the closest centroid to each data point in $D$. Hence, if we can show that there exists a vector $v \in \mathbb{R}^d$ such that the second term of Equation (1) is equal to zero (given $D$ and $W$) then we have shown what we wanted to prove. Since $|Y_{j\cdot}| = 1$ (every point is assigned to exactly one class), the matrix $Y^\top Y$ is a diagonal matrix, having the number of data points assigned to each class on the diagonal: $Y^\top Y = \mathrm{diag}(|Y_{\cdot 1}|, \ldots, |Y_{\cdot c}|)$. Hence, the trace term on the right of Equation (1) equals

$$\sum_{k=1}^{c} (2\mathbf{b}\mathbf{1}_c^\top - Z^\top Z)_{kk} |Y_{\cdot k}| = \sum_{k=1}^{c} (2b_k - \|W_{\cdot k}\|^2 - 2v^\top W_{\cdot k})|Y_{\cdot k}| - \|\mathbf{v}\|^2 m \qquad (2)$$

Full derivation of this equivalence is given in Appendix B. We define the vector $\mathbf{u} \in \mathbb{R}^c$ such that $u_k = b_k - \frac{1}{2}\|W_{\cdot k}\|^2$. The right term of Equation (2) is constant for a vector $\mathbf{v}$ satisfying $u_k = \mathbf{v}^\top W_{\cdot k}$ for $1 \le k \le c$. That is, we need to solve the following equation for $\mathbf{v}$:

$$\mathbf{u} = W^\top \mathbf{v} = V\Sigma U^\top \mathbf{v}.$$

Since the rank of $W$ is $c$ (full column rank), this equation has a solution. It is given by the SVD of $W = U\Sigma V^\top$, where $U \in \mathbb{R}^{d \times c}$ is a left orthogonal matrix ($U^\top U = I$), $\Sigma \in \mathbb{R}_+^{c \times c}$ is a diagonal matrix having only positive values, and $V \in \mathbb{R}^{c \times c}$ is an orthogonal matrix ($V^\top V = VV^\top = I$). Setting $\mathbf{v} = U\Sigma^{-1}V^\top \mathbf{u}$, this vector solves the equation. $\qquad \square$

The objective $\min_Y \|D - YZ^\top\|^2$ is the $k$-means cluster assignment objective in matrix factorization form. Hence, class predictions correspond to a Voronoi tesselation of $\mathbb{R}^d$. Classification accuracy is unaffected by the distance of the points $\phi(x)$ to the centroid, as long as they are in the correct Voronoi cell. Softmax confidence is high for points maximizing the inner product $\phi(x)^\top Z_{\cdot k}$, where $Z_{\cdot k}$ is the center of the predicted class, since

$$\mathrm{softmax}(W^\top \phi(\mathbf{x}) + \mathbf{b})_k = \frac{\exp(\phi(\mathbf{x})^\top W_{\cdot k} + b_k)}{\sum_{l=1}^{c} \exp(\phi(\mathbf{x})^\top W_{\cdot l} + b_l)} \cdot \frac{\exp(\phi(\mathbf{x})^\top \mathbf{v})}{\exp(\phi(\mathbf{x})^\top \mathbf{v})}$$

$$= \frac{\exp(\phi(\mathbf{x})^\top Z_{\cdot k} + b_k)}{\sum_{l=1}^{c} \exp(\phi(\mathbf{x})^\top Z_{\cdot l} + b_l)} = \mathrm{softmax}(Z^\top \phi(\mathbf{x}) + \mathbf{b})_k.$$

Hence, softmax confidence is high for points $\phi(\mathbf{x})$ aligning with the direction of their class center $Z_{\cdot k}$ and having a large norm in the transformed feature space. However, the empirical observation of neural collapse indicates that networks do in fact not map the training and test data arbitrarily far away from their centroids but rather into their neighborhood. Thus, one could attach semantic meaning to the distance of transformed samples to class centroids.

## 3.2 A Centroid-Based Robustness Certificate

In addition to the potential reflection of learned representations of a neural network, the distance of points to their centroid also determines the robustness of a network, as the following result shows (illustrated in Figure 2b).

**Theorem 2.** *Let $\mathbf{x} \in \mathbb{R}^n$ be a data point with predicted class $k$ and let $Z \in \mathbb{R}^{d \times c}$ be the centroid matrix. Assume $\phi : \mathbb{R}^n \to \mathbb{R}^d$ is Lipschitz continuous with modulus $L_\phi$. Any distortion $\Delta \in \mathbb{R}^n$ changing the prediction of point $\mathbf{x} + \Delta$ to another class $l \neq k$ has then a minimum size of*

$$\|\Delta\| \ge \frac{\|Z_{\cdot l} - Z_{\cdot k}\| - \|\phi(\mathbf{x} + \Delta) - Z_{\cdot l}\| - \|\phi(\mathbf{x}) - Z_{\cdot k}\|}{L_\phi}.$$

*Proof.* Let $\mathbf{x}, \Delta$ and $Z$ be as described above. We derive from the triangle inequality and from the Lipschitz continuity the following inequality:

$$\|\phi(\mathbf{x} + \Delta) - Z_{\cdot k}\| \le \|\phi(\mathbf{x} + \Delta) - \phi(\mathbf{x})\| + \|\phi(\mathbf{x}) - Z_{\cdot k}\|$$
$$\le L_\phi \|\Delta\| + \|\phi(\mathbf{x}) - Z_{\cdot k}\|. \qquad (3)$$

The triangle inequality also yields:

$$\|Z_{\cdot l} - Z_{\cdot k}\| \le \|\phi(\mathbf{x} + \Delta) - Z_{\cdot l}\| + \|\phi(\mathbf{x} + \Delta) - Z_{\cdot k}\|.$$

Subtracting $\|\phi(\mathbf{x} + \Delta) - Z_{\cdot l}\|$ yields a lower bound on the distance $\|\phi(\mathbf{x}) - Z_{\cdot k}\|$, which we apply in Equation (3) to obtain the final bound on the distortion $\Delta$. $\qquad \square$

Theorem 2 provides three explanations for the phenomenon that predictions of neural networks can be flipped by small perturbations: 1) the Lipschitz modulus is large, 2) class centroids are close to each other, and 3) point $\mathbf{x}$ or $\mathbf{x} + \Delta$ is not mapped close to their class centroid. Case 1 remains an open problem: this paper will not provide a manner to suitably control the Lipschitz constant. Case 2 motivates a maximum margin approach where the centroids are supposed to be as far away from each other as possible. Case 3 motivates a novel confidence score, which reflects the distance to the centroid. In the desirable situation where the Lipschitz constant is small and centroids are far away from each other, small perturbations only result in a prediction change if at least one of the points $\mathbf{x}$ or $\mathbf{x} + \Delta$ is far from its centroid. Hence, a confidence score reflecting the distance to the centroid is in alignment with the theoretical robustness guarantee of Theorem 2. Such a confidence score expresses not only the model's uncertainties, but also its vulnerability to attacks.

## 4 Gauss Confidence and Gauss Networks

Theorem 2 motivates the direct optimization for *good* centroids which are in current networks only indirectly learned. To do so, we employ a natural choice for a confidence measure which reflects the proximity to the cluster centroids via the Gaussian RBF-kernel function.

**Definition 1.** Given a function $\phi : \mathbb{R}^n \to \mathbb{R}^d$ and a centroid matrix $Z \in \mathbb{R}^{d \times c}$, we define predictions with *Gauss confidence* by the function $f_{\mathrm{ga}}(\mathbf{x})$, where for $k \in \{1, \ldots, c\}$

$$f_{\mathrm{ga}}(\mathbf{x}; \gamma)_k = \exp(-\gamma_k \|\phi(\mathbf{x}) - Z_{\cdot k}\|^2) \in (0, 1].$$

The parameter $\gamma \in \mathbb{R}_+^c$ determines how close the transformed samples have to be to their centroid in order to get a high confidence. The Gauss confidence is a special case of the multivariate Gaussian confidence (used, e.g., by DUQ networks) where the covariance matrix for class $k$ is equal to $\Sigma_k = 1/\gamma_k I$. This might seem restrictive, but this choice has two advantages. First, the margin between classes is more easily controllable over the distance between centroids when using Gauss confidence. For example, we can see in Figure 1 that the centroids of class 2 and 3 are comparatively far away, but due to the multivariate confidence, the margin between the confidently assigned areas is small. Second, the required storage for the covariance matrix in multivariate confidences increases vastly with the dimensionality of the penultimate layer space. For example, for a ResNet-50 architecture, the covariance matrix has a dimensionality of $2048 \times 2048$. We can also not decrease the required storage by employing a low rank approximation of $\Sigma^{-1}$, since this would introduce directions in which the confidence is always equal to one. In turn, the Gauss confidence requires only to store the $c$-dimensional vector $\gamma$.

### 4.1 Training Gauss Networks

We replace softmax with the Gauss confidence. While the weights $W$ of traditional networks indirectly determine the class centroids $Z$, our network learns the centroids directly, represented by the matrix $W = Z$, connecting the penultimate with the last layer.

Based on the result of Theorem 2, we aim for robust networks which map well-classifiable points close to the corresponding centroid and points which are difficult to classify further away from all centroids. In terms of clustering, we aim to minimize the within-class-scatter while maximizing the between-class-scatter. We propose a novel loss function that is based on the Gauss confidence and achieves both: it optimizes for feature transformations that map samples close to their centroid and centroids far away from each other. To this end, we introduce a parameter vector $\delta \in [0.01, 0.9]^c$.

$$\ell_{\mathrm{ga}}(f_{\mathrm{ga}}, Y^*) = -\frac{1}{m} \sum_{j=1}^{m} \log \left( \frac{f_{\mathrm{ga}}(\mathbf{x}_j; \gamma)_{y_j}}{f_{\mathrm{ga}}(\mathbf{x}_j; \gamma)_{y_j} + \sum_{k \neq y_j} f_{\mathrm{ga}}(\mathbf{x}_j; \delta \circ \gamma)_k} \right) + \log \left( f_{\mathrm{ga}}(\mathbf{x}_j; \gamma \circ \delta) \right)_{y_j}$$

The $\circ$ product indicates here the element-wise (Hadamard) product. The left term of the loss resembles cross-entropy loss. We introduce here the vector $\delta$ to simulate higher confidences

Table 1: Test accuracy (acc) and average confidence (conf) of MNIST, Cifar-10, Cifar-100 and ImageNet for softmax, softmax with adversarial training (+AT), DUQ and the proposed Gauss networks, using LeNet and ResNet (He et al., 2015) architectures.

| | MNIST LeNet | | Cifar-10 ResNet18 | | Cifar-100 ResNet18 | | ImageNet ResNet50 | |
|---|---|---|---|---|---|---|---|---|
| Network | acc↑ | conf | acc↑ | conf | acc↑ | conf | acc↑ | conf |
| Softmax | 99.3% | 0.99 | 94.4% | 0.98 | 76.4% | 0.86 | 75.7% | 0.81 |
| Softmax+AT | 99.0% | 0.99 | 79.7% | 0.63 | 56.8% | 0.50 | - | - |
| DUQ | 98.4% | 0.87 | 94.3% | 0.94 | 2.6% | 0.35 | - | - |
| Gauss | 99.3% | 0.93 | 94.3% | 0.91 | 75.8% | 0.64 | 73.8% | 0.65 |

for the wrong classes during training. This pushes centroids further away from samples that do not belong to their class, and hence increases the margin. The second term resembles negative log likelihood and increases the sample density around the centroids. The inclusion of $\delta$ forces the network to have higher sample density (and hence a smaller confidence ball around the centroid) when $\delta$ is large, and hence the margin around that class is supposedly small. To optimize for small values of $\delta$, we further employ a weight decay. We optimize the parameters of the network, as well as the parameters $\gamma$ and $\delta$, with SGD.

While the cross-entropy term pushes towards perfect training data classification, our proposed $\ell_{\mathrm{ga}}$ loss is more sensitive to the initialization than softmax models. We make use of the effect of neural collapse and our proven connection of softmax networks and nearest-centroid classifiers, and employ a warm start: we train a softmax network as usual for some epochs, then we compute the class centroids and determine the initial centroid matrix for the Gauss confidence. We then train the Gauss network for the remaining epochs with $\ell_{\mathrm{ga}}$.

## 5 EXPERIMENTAL RESULTS

We evaluate the proposed Gauss networks with respect to robustness and a suitable reflection of classification confidences by the Gauss confidence. For this purpose, we compare popular network architectures with the refined Gauss variant on MNIST (Lecun et al., 1998), Cifar-10 and -100 (Krizhevsky, 2009), and ImageNet (Russakovsky et al., 2015). As competitors, we choose Softmax networks (with a linear classifier and softmax confidence), DUQ models, and adversarially trained (AT) models (Madry et al., 2018). We also compare with respect to OOD detection with the method DDU, where we train a Gaussian mixture layer on top of the Softmax model to obtain an additional uncertainty measure. Our Pytorch implementation with scripts for training and attacking is publicly available[2].

### 5.1 CLASSIFICATION PERFORMANCE

We evaluate the feasibility of our proposed loss function to attain similar test set accuracies as softmax models. Parameter settings of our experiments are given in Appendix C.

Table 1 summarizes the performance results. For every dataset, we report the test accuracy and the average confidence of the predictions. Gauss networks achieve equivalent accuracies and similarly high confidences as softmax networks on MNIST and Cifar-10. On Cifar-100, we observe that the accuracy drops a bit but more notably, Gauss networks are less confident on these classes. While softmax networks are rather overconfident (the confidence is higher than the accuracy), Gauss networks tend to be less confident. This effect recurs on ImageNet: Gauss networks lose a few percentage points of accuracy w.r.t. softmax networks, but the latter have a higher confidence than accuracy. DUQ networks can compete on the MNIST and Cifar-10 datasets, but its training of Cifar-100 failed: the BCE loss pushes here the embedded training samples away from all centroids. In addition, the employed gradient penalty of DUQ networks quadruples the runtime. Consequently, we didn't optimize DUQ for ImageNet. Note that the runtime for one epoch is the same for Gauss and softmax

---

[2]https://anonymous.4open.science/r/GaussNetworks-A7E7

Table 2: Results of Attacks on MNIST and Cifar-10. We report the rate of successful attacks (rate) and the average confidence of the adversarial samples (conf) for each attack. The displayed attacks are Carlini & Wagner (C&W) (Carlini & Wagner, 2017), Fast Gradient Sign Attack (FGSM) (Goodfellow et al., 2014), Projected Gradient Descent (Madry et al., 2018) with $l_\infty$- ($l_\infty$PGD) and with $l_2$-norm ($l_2$PGD). For all attacks we set $\epsilon = 0.1$.

| Dataset | Network | C&W | | FGSM(0.1) | | $l_\infty$PGD(0.1) | | $l_2$ PGD(0.1) | |
|---|---|---|---|---|---|---|---|---|---|
| | | rate↓ | conf↓ | rate↓ | conf↓ | rate↓ | conf↓ | rate↓ | conf↓ |
| MNIST | Softmax | 39% | 0.81 | 12% | 0.75 | 23% | 0.80 | 1% | 0.54 |
| | Softmax+AT | 11% | 0.83 | 3% | 0.78 | 4% | 0.83 | 0% | 0.62 |
| | DUQ | 20% | 0.69 | 38% | 0.67 | 92% | 0.91 | 0% | 0.22 |
| | Gauss | 2% | 0.12 | 4% | 0.44 | 20% | 0.52 | 0% | 0.00 |
| Cifar-10 | Softmax | 93% | 0.84 | 76% | 0.84 | 93% | 1.00 | 40% | 0.96 |
| | Softmax+AT | 81% | 0.63 | 59% | 0.44 | 78% | 0.70 | 3% | 0.36 |
| | DUQ | 97% | 0.73 | 0% | 0.00 | 26% | 0.67 | 44% | 0.89 |
| | Gauss | 9% | 0.44 | 43% | 0.66 | 88% | 0.72 | 21% | 0.89 |

nets. We observe that adversarial training comes at a cost. Accuracy for AT on MNIST is comparable to its competitors, but drops by 14 and 20 percentage points on Cifar-10 and Cifar-100, respectively, when compared to the plain Softmax model.

## 5.2 Robustness to Attacks

We evaluate the robustness of models by means of four attack methods: Carlini & Wagner (C&W) (Carlini & Wagner, 2017), Fast Gradient Sign Attack (FGSM) (Goodfellow et al., 2014), Projected Gradient Descent (Madry et al., 2018) with the $l_\infty$ ($l_\infty$PGD) and $l_2$ ($l_2$PGD) norms. For our attacks, we employ the advertorch library (Ding et al., 2019). One of the theoretical strengths of the Gauss and DUQ confidence is the possibility to identify outliers. Since the softmax confidence of the predicted class is always larger than $\frac{1}{c}$, we also specify that a point is considered as an outlier if the confidence of the predicted class is smaller than $\frac{1}{c}$: if we have ten classes, then the outlier threshold is 0.1. We train all networks with the same number of epochs.

We investigate to which extent Gauss networks provide an inbuilt robustness (cf. Theorem 2) on the MNIST and Cifar-10 datasets. Table 2 reports the fraction of images for which the attacks lead to misclassification, and the average confidence of the network for these misclassifications. For both measures, lower is better. For all attacks on both datasets, the Gauss network outperforms the softmax network in terms of both attack success rate and confidence in the adversarial examples. Gauss networks are less or equally susceptible to C&W and FGSM(0.1) attacks than the adversarially trained network. Notably, the AT models are trained to defend against the $\ell_\infty$PGD attack, hence it comes at no surprise that AT models are more robust to PGD attacks than Gauss networks. DUQ is sometimes more, sometimes less robust than the softmax model. On Cifar-10, DUQ is notably more robust to FGSM and $\ell_\infty$PGD attacks. Here, the attacks seem to overshoot: decreasing the allowed perturbation size actually increases the attack success rate (cf. Appendix E). A qualitative analysis of the adversarial examples is given in Appendix D.

## 5.3 OOD Detection

Ideally, good confidence measures are able to reflect what the model knows, i.e., what it has been trained on. This reflection of areas of confidence can be used for OOD detection. Figure 3 displays the Gauss confidence against three uncertainty measures on Fashion MNIST trained networks with MNIST being the OOD data (top) and on Cifar-10 trained models with SVHN (Netzer et al., 2011) being the OOD data (bottom). We train Fashion MNIST on the LeNet architecture and Cifar-10 on ResNet18. Our competitors are softmax with entropy uncertainty (high entropy indicates high uncertainty), DDU (Mukhoti et al., 2021) using the log of the sum of the Gaussian mixture pdfs as a certainty measure, and

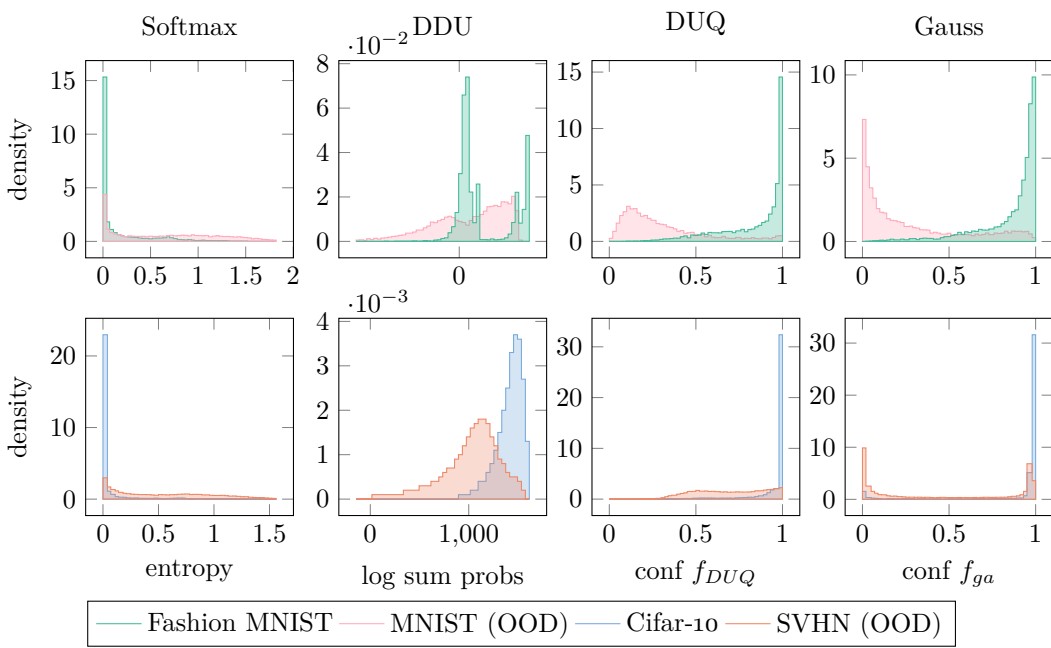

Figure 3: Histograms of uncertainty measures and confidences for models trained on Fashion MNIST (top) and Cifar-10 (bottom), deployed also on out-of-distribution data.

DUQ networks' confidence. We train the DDU softmax model as recommended with spectral normalization on the ResNet-18 architecture, using a soft Lipschitz bound of 3. The plots indicate that no measure delivers a perfect split between OOD and in-distribution data. However, we can observe a distinguishing characteristic of Gauss networks: using a threshold of 0.5 confidence separates the majority of OOD samples from the majority of in-distribution samples. For all other uncertainty measures, the cut-off value that separates the distributions is less consistently evident.

## 6 Conclusions

We introduce a loss function to optimize for neural networks with Gauss confidence, a centroidal confidence measure for Deep Neural Networks, instead of the standard softmax classification and the recently proposed alternative of DUQ networks (van Amersfoort et al., 2020). We prove that softmax-based classifiers are nearest centroid classifiers (cf. Theorem 1). Softmax classification accuracy is typically insensitive to the distance of the points to its centroid: as long as a point lies in the correct Voronoi cell, softmax is confident of its decision. We prove that one can certify the robustness of the network through these distances (cf. Theorem 2). We build on this centroid-based robustness certificate to define our Gauss confidence (cf. Definition 1). Its islands of confidence, illustrated in the rightmost plot of Figure 1, are in contrast to the wider confidence regions generated by the competitors.

Gauss networks can achieve comparable accuracies to softmax networks (cf. Table 1), but are more robust against adversarial attacks (cf. Table 2). When the softmax networks are enhanced with adversarial training, they are more robust against PGD attacks than Gauss networks, but Gauss networks are more robust against C&W and FGSM attacks and achieve better accuracies. In some cases, DUQ networks match Gauss networks in their robustness (though Gauss networks are more robust on the Cifar-10 dataset), but DUQ networks cannot achieve the same accuracy as Gauss or softmax networks when the dataset encompasses $\gg 10$ classes. In future work, we intend to improve Gauss networks by allowing for multiple centroids per class. This would allow for classes that are not one monolithic whole: they could consist of several parallel subsets, or even hierarchical concepts.

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

## A    Derivation of Equation (1)

We employ the fact that the $m$-dimensional constant one vector is equal to $\mathbf{1}_m = Y\mathbf{1}_c$, because $Y_{j\cdot}\mathbf{1}_c = |Y_{j\cdot}| = 1$. Using this relation, we obtain Equation (4).

$$\arg\max_Y \operatorname{tr}(Y(W^\top D^\top + b\mathbf{1}_m^\top))$$

$$= \arg\max_Y \operatorname{tr}(YW^\top D^\top) + \operatorname{tr}(Yb\mathbf{1}_c^\top Y^\top)) \tag{4}$$

$$= \arg\min_Y \|D\|^2 - 2\operatorname{tr}(YW^\top D^\top) - 2\operatorname{tr}(\mathbf{1}_m v^\top D^\top) - \operatorname{tr}(2b\mathbf{1}_c^\top Y^\top Y)) \tag{5}$$

$$= \arg\min_Y \|D\|^2 - 2\operatorname{tr}(Y(W^\top + \mathbf{1}_c v^\top)D^\top) - \operatorname{tr}(2b\mathbf{1}_c^\top Y^\top Y)) \tag{6}$$

$$= \arg\min_Y \|D\|^2 - 2\operatorname{tr}(YZ^\top D^\top) + \operatorname{tr}(YZ^\top ZY^\top) - \operatorname{tr}(YZ^\top ZY^\top) - \operatorname{tr}(2b\mathbf{1}_c^\top Y^\top Y)) \tag{7}$$

$$= \arg\min_Y \|D - YZ^\top\|^2 - \operatorname{tr}((2b\mathbf{1}_c^\top + Z^\top Z)Y^\top Y)) \tag{8}$$

In Equation (5) we change the maximization to the minimization of the negative objective function times two. Furthermore, we add the constant terms $\|D\|^2$ and $-2\operatorname{tr}(\mathbf{1}_m v^\top D^\top)$, that do not change the optimizer $Y$. Using again the fact that $\mathbf{1}_m = Y\mathbf{1}_c$, we can write the two terms in the middle of Equation (5) as one in Equation (6). In Equation (7), we complete the square, arriving at the objective of Equation (8).

## B    Derivation of Equation (2)

The diagonal elements of the matrix $2b\mathbf{1}_c^\top + Z^\top Z$ are given by $2b_k + Z_{\cdot k}^\top Z_{\cdot k}$, which yields Equation (9). We then insert the definition of $Z_{\cdot k} = W_{\cdot k} + v$ in Equation (10), expand the term and simplify $\sum_{k=1}^c |Y_{\cdot k}| = m$, arriving at Equation (11).

$$\sum_{k=1}^c (2b\mathbf{1}_c^\top + Z^\top Z)_{kk}|Y_{\cdot k}| = \sum_{k=1}^c (2b_k + Z_{\cdot k}^\top Z_{\cdot k})|Y_{\cdot k}| \tag{9}$$

$$= \sum_{k=1}^c (2b_k + (W_{\cdot k} + v)^\top (W_{\cdot k} + v))|Y_{\cdot k}| \tag{10}$$

$$= \sum_{k=1}^c (2b_k + \|W_{\cdot k}\|^2 + 2v^\top W_{\cdot k} + \|v\|^2)|Y_{\cdot k}| \tag{}$$

$$= \sum_{k=1}^c (2b_k + \|W_{\cdot k}\|^2 + 2v^\top W_{\cdot k})|Y_{\cdot k}| + \|v\|^2 m \tag{11}$$

## C    Experimental Setup

We employ for Gauss networks a warmup of 10 epochs on MNIST and Cifar-10, 30 epochs on Cifar-100, and 60 epochs on ImageNet. MNIST is trained for 60 epochs, while the other datasets are trained for 100 epochs (including the warmup). For softmax and Gauss networks we employ a similar learning rate schedule, and we use the advised learning rate schedule for DUQ networks (as provided for Fashion MNIST and Cifar-10). We train the LeNet architecture with a learning rate of 0.1 for softmax nets and 0.02 for Gauss nets, which is decayed by a factor of 10 at epoch 40. We train the ResNet-18 architecture with a learning rate of 0.1 for softmax nets and 0.02 for Gauss nets, which is then decayed by a factor of 10 at epoch 60 and 80. The AT models are trained with the $l_\infty$ PGD attack. We employ a standard setting with 10 iterations in the inner loop and $\epsilon = 0.1$ for MNIST and $\epsilon = \frac{8}{255}$ for CIFAR-10/100.

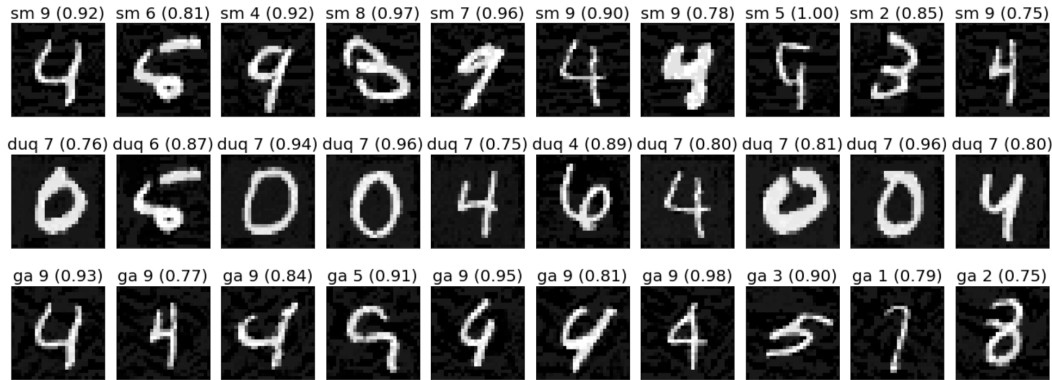

Figure 4: Adversarial examples achieved with the $l_\infty$ PGD attack on MNIST with confidence $\geq 0.7$. Above each image you find the network (sm: softmax, duq: DUQ, ga: Gauss), the erroneously predicted class, and the confidence of the prediction in parentheses.

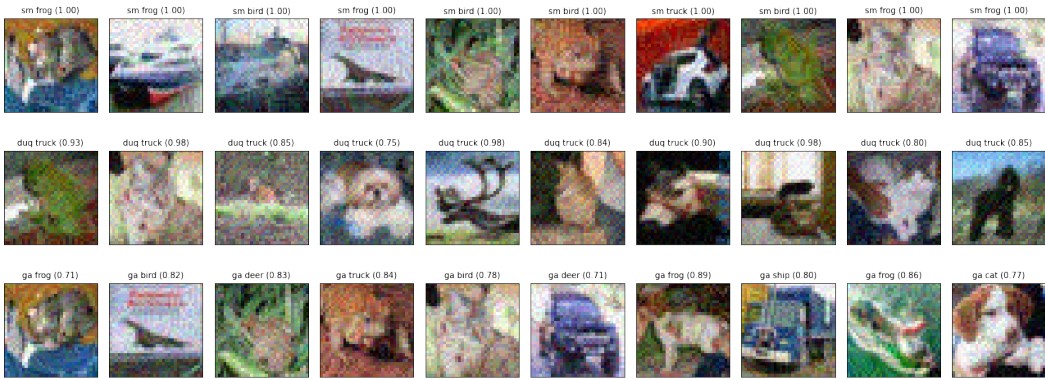

Figure 5: Adversarial examples achieved with the $\ell_\infty$ attack on Cifar-10 with confidence $\geq 0.7$. Above each image you find the network (sm: softmax, duq: DUQ, ga: Gauss), the erroneously predicted class, and the confidence of the prediction in parentheses.

## D    Cifar-10 Adversarial Examples

The most successful Gauss network attacks are the $l_\infty$ PGD attack on MNIST and Cifar-10. Confidently assigned adversarial examples are displayed in Figures 4 and 5. Adversarial examples on MNIST indicate that Gauss networks often confuse a four with a nine; these also are the classes with the nearest centroids (distance of 9.5; average centroid distance is $12.1 \pm 1.4$). The adversarial examples on Cifar-10 are less conclusive. While some of the adversarial examples for Gauss net are justifiable, other adversarial examples hardly make sense. This indicates that our proposed confidence measure does not yet provide a definite safeguard against learning shortcuts, but it provides a first step towards indications of model representations and knowledge.

## E    Effect of the Parameter $\epsilon$ in Attacks

We plot in Figure 6 the attack success rate against the parameter $\epsilon$ that determines the bound on the perturbation size. As expected, the higher the parameter $\epsilon$, the higher the attack success rate for softmax and Gauss networks. In contrast, we observe that DUQ models are for the FGSM and $\ell_\infty$ PGD attacks more susceptible to small perturbations of the input. This observation suggests that those attacks possibly overshoot, such that larger perturbations result in images that are mapped outside of the confidently assigned region.

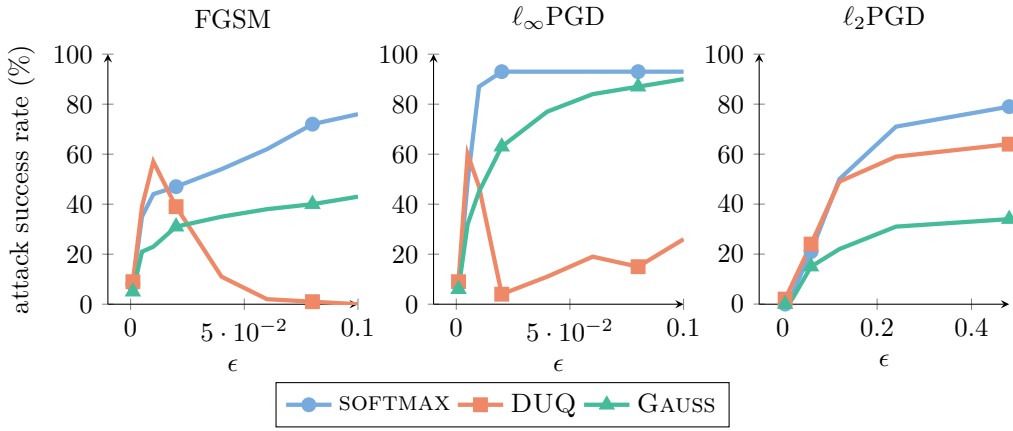

Figure 6: Variation of parameter $\epsilon$, comparison of attack success rates (the lower the better) on Cifar-10 trained networks for the FGSM, $\ell_\infty$PGD and $\ell_2$PGD attack.

In comparison, the empirical analysis of the effect of $\epsilon$ further indicates that the proposed Gauss networks are indeed well protected against small perturbations in the input.

