# OpenReview forum: "Islands of Confidence: Robust Neural Network Classification with Uncertainty Quantification"
_ICLR.cc/2023/Conference — Submitted to ICLR 2023_

### Official Review · Reviewer_gCc4 · 2022-10-23

**Confidence:** 3
**Clarity, Quality, Novelty And Reproducibility:** The paper is well written and easy to…
**Correctness:** 3
**Technical Novelty And Significance:** 3
**Empirical Novelty And Significance:** 3
**Recommendation:** 5

**Strength And Weaknesses:**

Strength:

* A interesting work that improves the single-model uncertainty methods based on RBF-style classifier (i.e., DUQ).
* Author investigated (both theoretically and empirically) the implication of adversarial robustness of the distance-based approach.

Weakness:

* Performance is lacking on large-scale benchmarks. There are several single-model methods developed and tested on, e.g., ImageNet with ResNet50 (e.g., see MIMO or SNGP on Uncertainty Baselines: https://github.com/google/uncertainty-baselines/tree/main/baselines/imagenet), which are generally competitive or outperform standard softmax classifier in Table 1. This may somehow discount the emprirical utility of Gauss classifier, as it still incurs a hard uncertainty v.s. utility tradeoff. This somehow leads me to question the utility of centroid-based classifier in more difficult tasks (CIFAR-100 and ImageNet).

* Missing evaluation metric for uncertainty performance: In Table 1, please consider evaluate model uncertainty using a standard evaluation metric such as ECE.

* Missing treatment of Lipschitz constant control / related literature review. As authors commented, Lipschitz constant is important for guanrantee model's uncertainty and robustness performance. However, there are a stream work in the single-model uncertainty literature (e.g., [SNGP](https://arxiv.org/abs/2205.00403), [DUE](https://arxiv.org/abs/2102.11409), [DDU](http://www.gatsby.ucl.ac.uk/~balaji/udl2021/accepted-papers/UDL2021-paper-022.pdf) etc) that tackles this by combining spectral normalization with distance-based classifier, which generally outperforms DUE and works well for high-dimensional benchmarks. Please consider include them into related work, and optionally absorbing some of these baseline techniques into methodology or include these previous work as baselines.


**Summary Of The Paper:**

This work proposes modeling the predictive confidence of the DNN model using RBF-style output layers (Definition 1). This choice is justified by a theoretical argument on the connection between softmax classifier and the  neares centroid classifier, for which a robustness guanrantee can be derived (Section 3). The paper then conducted vision experiments based on standard benchmark and architecture (ResNet-18/50 on MNIST/CIFAR/ImageNet), ans shows accuracy improvement over DUQ, and adversarial robustness improvement over softmax classifier.

**Summary Of The Review:**

This is an interesting work that theoretically establishes the connection between softmax classifier and centroid classifier, empirically proposed an improved formulation of RBF-style last-layer classifier than outperforms its similar predecessors (i.e., DUQ), and brings the consideration of adversarial robustness into the picture.

However, the work is slightly disconnected from the current uncertainty literature in that it does not sufficiently review or compare with its contemporary methods (e.g., SNGP, DUE) that handles Lipschitz control and uses distance-based classifier, and empirically its accuracy performance is not on par with the other uncertainty methods, and its uncertainty performance is not rigorously evaluated with standard performance metrics (e.g., ECE). Adding literature review and additional baseline comparison should address these issues to some extent.

---

> ### Author Response · Authors · 2022-11-19
> **Individual Comments**
>
> Thank you very much for your comments. We would like to respond mainly to the comment about performance. We feel that we already addressed the other raised points in our response to reviewer SMCi, and would kindly like to ask you to review Points 3 and 4 in the response. Many thanks for pointing out the related models, we have integrated a comparison with DDU, as it fits to our Gaussian distribution-based approach.
>
> ### 1. Lacking Performance on large scale benchmarks
> Indeed, Gauss nets achieve on Cifar-100 and Imagenet ~2\% lower accuracy. While achieving higher accuracy would be always nicer, we do not really see the point in optimizing for these benchmarks. Both datasets have obvious flaws. Cifar 100 has such a low resolution that it is often not even for humans possible to distinguish between the classes. Imagenet has a big portion of mislabeled classes (https://arxiv.org/abs/2208.11695), and although often multiple objects appear in one picture, only one class is considered as the correct one. We do not think that achieving the highest accuracy is the main goal. We hope that we could show that Gauss nets bring other practically relevant properties to the table, such as robustness.
>
> In addition, we would like to point out that we only use 100 epochs to train our models, while the listed competitors use twice as much. Certainly with time, effort and compute, it would be also possible to train Gauss Nets towards higher accuracies.

---

### Official Review · Reviewer_D1u1 · 2022-10-24

**Confidence:** 3
**Correctness:** 2
**Technical Novelty And Significance:** 3
**Empirical Novelty And Significance:** 2
**Recommendation:** 5

**Clarity, Quality, Novelty And Reproducibility:**

This paper is well-presented and organized. The proposed idea is novel. The authors provide the code for experiment result reproduction.


**Strength And Weaknesses:**

Strengths:

1. The authors formally prove that independent of optimization-procedural effects, a set of centroids always exists such that softmax classifiers are the nearest centroid classifiers.
2. The authors proposed a new confidence measure that is centroid-based and hence no longer suffers from the artificial confidence inflation of out-of-distribution samples.
3. The proposed centroidal confidence measure provides a robustness certificate against attacks.


Weaknesses:

1. What is the motivation of Section 3? What is the connection between uncertainty and robustness?
2. Lack of important related work of uncertainty quantification, such as [1, 2].
3. It is better to add more baselines to demonstrate the effectiveness of the proposed method.
4. Since the paper focuses on uncertainty quantification, it may necessary to show some experiments result about out-of-distribution detection.


[1] Malinin, Andrey, and Mark Gales. "Predictive uncertainty estimation via prior networks." Advances in neural information processing systems 31 (2018).

[2] Malinin, Andrey, and Mark Gales. "Reverse kl-divergence training of prior networks: Improved uncertainty and adversarial robustness." Advances in Neural Information Processing Systems 32 (2019).


**Summary Of The Paper:**

This paper focuses on uncertainty quantification and robustness. The authors find that existing work in uncertainty quantification mostly revolves around the confidence reflected in the input feature space. The authors focus on the learned representation of the network and analyze the confidence in the penultimate layer space. The proposed new confidence measure is centroid-based and hence no longer suffers from the artificial confidence inflation of out-of-distribution samples.


**Summary Of The Review:**

See Strength And Weaknesses:

------------ after rebuttal ------------

The authors' response addressed some of my concerns. But I still suggest adding more baselines and out-of-distribution detection experiments in the future version to demonstrate the effectiveness of the proposed method. Therefore, I will keep my score.

---

> ### Author Response · Authors · 2022-11-19
> **Individual Comments**
>
> Thank you very much for your comments. Let us respond to weakness 1, we integrated your other raised points in the revision. Many thanks for pointing out the additional literature.
>
> ### What is the motivation of Section 3? What is the connection between uncertainty and robustness?
> Our theoretical result, showing that softmax classifiers are also centroid classifiers, indicates that we will not lose expressivity when we transfer from softmax models with a linear classifier to a centroid classifier. That is, in theory, nearest centroid classifiers are just as powerful as softmax classifiers. Further, it motivates to apply the optimized training procedures for softmax models and to use these models as initialization/warm start for a centroid-based classifier. Our intuition is that the transformed feature space (penultimate layer space) conveys information about the semantics of learned representations, which could be more clearly extracted by a centroid-based confidence measure.
>
> The connection between uncertainty and robustness is for Gauss nets given by the fact that confidently classified clean inputs are harder to perturb locally such that the perturbed sample is also confidently assigned to another class (cf. Figure 2b). Low confidence indicates here  a potential security risk: the sample might be susceptible to attacks or be the result of an unsuccessful attack itself.

---

### Official Review · Reviewer_SMCi · 2022-10-28

**Confidence:** 4
**Clarity, Quality, Novelty And Reproducibility:** The paper is clear and focused; the s…
**Correctness:** 3
**Technical Novelty And Significance:** 2
**Empirical Novelty And Significance:** 2
**Recommendation:** 5

**Strength And Weaknesses:**

Strengths
* The paper is well-written, the explanation is clear and the flow is good.
* The proposed method appears naturally, backed up by a sound theoretical background.
* Though it's closely related to another method, the one proposed in the paper feels simpler, requiring fewer resources and being more robust at the same time.

Weakness
My main concern is that experiments feel shallow
* First of all, the authors refer to DUQ and make comparisons mostly with it. But DUQ by itself focuses on out-of-distribution detection, not robustness, so I would expect out-of-distribution experiments. If the robustness is expected to be the strongest point for the method, then I would expect some baseline from this area, e.g. adversarial training or improved Lipschitz constant with methods like spectral normalization.
* The authors mention restricting of Lipschitz constant and it's a part of a proof condition, but it's not addressed in the implementation and experiments. What's more concerning, local Lipschitz constant restriction with spectral normalization is part of the DUQ method, but I feel that it wasn't used in the paper this way. The provided repository doesn't contain the code for it, so I have some concerns about the validity of the provided results.
* I'm not sure what conclusion should we make from confidence in table 1. Mean confidence is not a very meaningful metric, as with postprocessing like temperature scaling you can get almost arbitrary values for it. The calibration plot would be much more meaningful in this case. Particularly, one can argue that having mean confidence 0.87 with 0.99 accuracy is strongly under-confident.
* While the statement in theorem 1 seems valid, there are a few issues in derivation, I believe it could use some polishing. E.g., missed opening bracket in eq.1. The sign for b seems wrong as well there, should be '-' in the third line of the appendix.  It's stated that the second term eq.1 could be made zero, but in fact, it is only shown to be constant.

**Summary Of The Paper:**

The authors propose a new method for measuring confidence and increasing the robustness of neural networks. The core of the method is replacing the last linear layer and softmax with learnable per-class Gaussians (centroids).
To address convergence problems, the paper proposes simplified Gaussians and adding negative log-likelihood with a margin to a loss.
The method is backed by proof that classification with centroids is at least as expressive as linear layer classification.

**Summary Of The Review:**

While I have concerns about the experimental part, the paper is coherent and original overall.

---

> ### Author Response · Authors · 2022-11-19
> **Individual Comments**
>
> Thank you very much for your comments. We would like to respond to the points you raised.
> ### 1. Is it fair to compare against DUQ with regard to robustness?
> We understand your point. Indeed DUQ was not particularly designed to be robust against attacks, but it does incorporate many of the building blocks that we theoretically identify as being important for robustness: a Lipschitz constant regularization and the fit of the confidence to the training data distribution in the latent space (which reduces the area of confidently assigned data points and therewith also the area that needs to be "hit" with an attack).
> However, we also agree that OOD detection experiments are useful to complete the picture, which you will find in our revision. Likewise, we added the suggested adversarially trained model as a robustness baseline.
> ### 2. Implementation of Lipschitz regularization in DUQ
> DUQ uses a Lipschitz regularization with gradient penalty (GP), which is implemented in the optimization.py file of our repository. We copied the gradient penalization code from the repository of DUQ and are confident that it is correctly implemented.
>
> We have also added a small paragraph in the revision (in Related Work), discussing the applicability of Lipschitz regularization in methods such as DUQ, DDU and SNGP. In short, those regularizations have to be watered down, because otherwise, the performance suffers notably. Further, the efficability of spectral normalization seems to depend also on the architecture. Methods such as DDU and SNGP show that the biggest effect is achieved with a wide resnet architecture. However, when we trained resnet-18 on Cifar10 with spectral normalization using the default parameters, we even observed a small decrease in robustness to attacks. Other characteristics, such as the AUROC curve of DDU for nets trained with or without spectral normalization, did not change notably.
> We tried using GP for Gauss nets, but it made the training slow and could not notably increase the robustness against attacks. Since bounding the Lipschitz constant in an effective manner is still such an unsolved problem, we decided to focus on increasing the margin instead, which is actually more easy to control.
> ### 3. The average confidences
> The average confidences for all networks are supposed to indicate that differences in the observed robustness are not just an effect of an overall lower confidence assignment. Because we say that an attack has failed when it has a confidence <1/num_classes, robustness could here also be achieved by assigning all points with low confidence to a class. The average confidences show that this is not the case. The mentioned underconfident model with mean confidence 0.87 and 0.99 accuracy is actually DUQ. We're sorry for the confusion but we mixed up the numbers in the latex file.
> ### 4. Proofs
>  Many thanks for checking the proofs! We polished the presentation in the revision.

---

> > ### Comment · Reviewer_SMCi · 2022-12-13
> > **Thanks for the comments**
> >
> > Dear authors,
> >
> > I would like to thank you for the comments. I tend to keep my scores but encourage you to work on the issues raised by me and other reviewers for the future submissions. In particular, more extensive OOD experiments are of interest.

---

### Author Response · Authors · 2022-11-19
**Revision of our Islands of Confidence paper**

Dear Reviewers,

Many thanks for your valuable input. We feel like you all are very well knowledgeable in the field and we’re happy to see your assessment of our paper to be interesting, novel, and well-written. As weakness you align in pointing out doubts regarding the experiments. Correspondingly, we have updated our manuscript and we hope that you find the time to look at our revision. Newly created text and baselines are marked in blue.

In our updated manuscript, we have added baselines and experiments. We added a standard OOD detection experiment, comparing Cifar-10 vs. SVHN and Fashion MNIST vs MNIST. As a competitor for the OOD detection methods, we add to the existing set of competitors the method DDU which fits a GMM layer to a traditional softmax network (as recommended, we use Spectral normalization to train the softmax model of DDU for the ResNet18 architecture, using the default parameter 3).

For the robustness experiments, we have added a network adversarially trained against the $\ell_\infty$PGD attack. Those experiments are quite nicely showing the trade-off between robustness and accuracy and how powerful our approach actually is in delivering both a high test-set accuracy and robustness.

Further, we have added all recommended literature to the body of related work, thank you for pointing them out.

We would also like to note that we recognized a small parameter mix-up in the employed attacks. We retrained everything to make sure that it is correct, but this is why some of the numbers are different in the revision. Not much has changed overall, mostly DUQ is actually less robust against attacks on MNIST and partially more robust on Cifar-10.

If you have any doubts about the implementation, feel free to look in our repository. We did our utmost best to make the experiments accessible and reproducible.

---

### Decision · Program_Chairs · 2023-01-20

**Decision:**

Reject

**Justification For Why Not Higher Score:**

There are several concerns with the paper, see reviews.

**Justification For Why Not Lower Score:**

N/A

**Metareview: Summary, Strengths And Weaknesses:**

The paper investigates uncertainty quantification in NNs by treating them as a softmax classifiers. The paper establishes a formal relationship between softmax and Voronoi region driven classification. The relationship between the two is obtained through multivariate Gaussian distribution. This relationship is then exploited to suggest Gaussian confidence scores, which can be seen as a measure of
uncertainty quantification and also robustness.

While there was appreciation of the proposed measure but the reviews express substantive concerns mostly related to relationship with past work. The rebuttal tried to address some of them but the current draft does not seem to be ready for a selective conference like ICLR

**Summary Of Ac-Reviewer Meeting:**

Not needed